# ATP5B Is an Essential Factor for Hepatitis B Virus Entry

**DOI:** 10.3390/ijms23179570

**Published:** 2022-08-24

**Authors:** Keiji Ueda, Yadarat Suwanmanee

**Affiliations:** Division of Virology, Department of Microbiology and Immunology, Graduate School of Medicine, Osaka University, 2-2 Yamada-oka, Suita 565-0871, Japan

**Keywords:** hepatitis B virus (HBV), ATP5B, attachment/entry

## Abstract

Elucidation of the factors responsible for hepatitis B virus (HBV) is extremely important in order to understand the viral life cycle and pathogenesis, and thereby explore potential anti-HBV drugs. The recent determination that sodium taurocholate co-transporting peptide (NTCP) is an essential molecule for the HBV entry into cells led to the development of an HBV infection system in vitro using a human hepatocellular carcinoma (HCC) cell line expressing NTCP; however, the precise mechanism of HBV entry is still largely unknown, and thus it may be necessary to elucidate all the molecules involved. Here, we identified ATP5B as another essential factor for HBV entry. ATP5B was expressed on the cell surface of the HCC cell lines and bound with myristoylated but not with non-myristoylated preS1 2-47, which supported the notion that ATP5B is involved in the HBV entry process. Knockdown of ATP5B in NTCP-expressing HepG2 cells, which allowed HBV infection, reduced HBV infectivity with less cccDNA formation. Taken together, these results strongly suggested that ATP5B is an essential factor for HBV entry into the cells.

## 1. Introduction

Hepatitis B virus (HBV), a member of the Hepadnaviridae family, is a small DNA virus with unusual features similar to retroviruses. At this point, HBV replicates through an RNA intermediate and can integrate into the host genome. The unique features of the HBV replication cycle confer a distinct ability of the virus to persist in infected cells [1].

Hepatitis B virus (HBV) is a life-threatening agent that causes acute and chronic hepatitis, liver cirrhosis, and liver cancer. It has been estimated that several million people are infected with the virus and are at risk of such diseases [2]. Although HBV was first identified more than a half-century ago [3,4], there had been none to allow HBV to infect in vitro, and thus there had been so few cases of conducting HBV experiments in vitro. Recently, sodium taurocholate co-transporting peptide (NTCP) was reported to be an essential factor for HBV entry [5]. The expression of NTCP in human-originated hepatocellular carcinoma (HCC) cell lines finally allowed the development of a cellular model of HBV infection [5]. This was the most important development for HBV study since, previously, HBV had been studied only with molecular techniques such as overexpression of the genes with or without mutation, and no system was available to elucidate the natural HBV life cycle in vitro. Although the determination of the centrality of NTCP to HBV entry improved our understanding of the HBV life cycle, many aspects of the HBV life cycle remain to be clarified. For example, it remains to be determined whether NTCP is the only factor required for HBV attachment and entry into the cells, or whether there are many other cofactors involved in HBV entry, as there are for the entry of other viruses such as human immunodeficiency virus 1 (HIV-1) and hepatitis C virus (HCV). In addition, if such factors exist, it would be important to determine whether they work independently of or dependently on NTCP.

Recently, ATP1B, a component of F_1_F_0_ ATP synthase, was reported to be involved in some viral life cycles by interacting with viral gene products [6,7]. Another Na^+^/K^+^ ATPase, ATP1B3, was found to modulate BST-2 mediated restriction of human immunodeficiency virus 1 [8], and conversely, to interact with the 3A protein of enterovirus 71 (EV71) and inhibit EV71 amplification by promoting type I interferon production [9]. ATP5B was also reported to be involved in the replication and/or life cycles of several viruses [10,11].

Our own group performed various experiments to identify factors that were required for HBV entry as preS1-binding factors, since the preS1 of the HBV large S protein had been reported to be a viral ligand for HBV entry. In the process, we found that ATP5B was one of the essential factors for HBV entry. ATP5B is a beta subunit of the F_1_ part of the F_1_F_0_ ATP synthase that is responsible for the majority of ATP synthesis in many organisms [12]. Though it seems to be mostly present in the inner membrane of mitochondria, ATP5B has also been shown to be expressed on cell surfaces, including in an HCC cell line (HepG2) [13,14,15,16].

In the present study, we identified ATP5B as a preS1-interacting factor by means of a preS1 peptide pull-down assay followed by an immunoblotting assay (Western blotting assay). Knockdown of ATP5B in an NTCP-expressing HepG2 cell line that allowed HBV infection resulted in a decrease in HBV infectivity. Finally, we found that ATP5B did not interact with NTCP physically, and thus, it should function as one of the molecules responsible for the HBV entry independent of NTCP.

## 2. Results

### 2.1. Identification of ATP5B as a Factor Interacting with preS1

We performed a pull-down experiment with myristoylated preS1 peptide: 2-47aa (myrPS1) or PS2: 2-55aa (PS2) using HepG2 or NTCP-expressing HepG2 cells (NTCP/G2). HBV large S envelope protein is myristoylated at the 2nd amino acidglycine next to the methionine, and it is functionally very important for the attachment and entry of HBV into the cells [17,18]. We found a band near 50 kd that was clearly stained with myrPS1 but not with PS2, and analyzed it by MALDI-TOF/MS; ATP5B was identified as a result (Appendix A). To confirm the interaction, the same pull-down followed by an immunoblotting assay was performed using HepG2 lysate. The results showed that cellular ATP5B was pulled down with myrPS1 but not PS2. We also tested whether ATP5A1 was also pulled down with the peptides, since ATP5A1 and ATP5B were in the same complex as an ATPase/synthase. Interestingly, ATP5A was not detected, suggesting that ATP5B was present independently of such a complex, or that, on binding with myrPS1, the complex was dissociated while the myrPS1–ATP5B interaction was preserved (Figure 1).

### 2.2. Myristoylation of PS1 Affects the Interaction

Myristoylation of PS1 at the N-terminal glycine residue was important for the interaction between PS1 and NTCP (sodium taurocholate cotransporting peptide), an essential receptor for HBV attachment/entry into hepatocytes [5]. Thus, we also analyzed the importance of myristoylation of the PS1 peptide using the non-myristoylated PS1 peptide (non-myrPS1) and several PS1 mutant peptides (myristoylated PS1 N9K (myrPS1 N9K) and myristoylated PS1 F11D (myrPS1 F11D)). As shown in Figure 1B, non-myrPS1 and myrPS1 N9K lost most of the binding activity with ATP5B, suggesting that myristoylation of PS1 and asparagine at the 9th amino acid in the preS1 region should be important for the binding, but not phenylalanine at the 11th amino acid. The requirement of myristoylation for the binding also suggests that ATP5B may be present on the cell surface and involved in the HBV attachment/entry process.

### 2.3. Colocalization of PS1 (a Large S Protein: LS) and ATP5B in the Cells

In order to test whether PS1 and ATP5B interacted in the cells, we performed an immunofluorescence assay (IFA) using HB611 cells that were producing HBV stably [19] and Huh6 cells that were parental cells of HB611 cells and did not produce HBV. Since ATP5B was mainly localized in mitochondria, a strong merged signal was observed in the cytoplasm (Figure 1C). Nevertheless, the signal was sufficient to show that PS1 and ATP5B existed in close proximity in vivo, although the assay was not good enough to reveal their interaction on the cell surface.

### 2.4. ATP5B Is Present on the Cell Surface

Some studies have reported that ATP5B was present on the cell surface [14,16,20,21,22]. In our case, ATP5B was expected to exist on the cell surface and to play some role in HBV attachment/entry. To confirm this, we labeled the cell surface with sulfo-NHS-LC-LC-biotin, which was not incorporated into the cell. Then, samples bound with streptavidin-Sepharose^®^ were separated on an SDS-PAGE and blotted with an anti-ATP5B antibody. The images of the original HepG2 cells as well as NTCP/G2 showed that ATP5B was almost certainly present on the cell surface but not GAPDH (Figure 2A,B). These data strongly suggested that at least a portion of the total cellular ATP5B should be present on the cell surface and bind with the preS1 region of HBV infectious particles for entry.

### 2.5. ATP5B Knockdown Loses HBV Infectivity

We next tried to establish ATP5B-knocked down (ATP5B KD) cells in order to test the importance of ATP5B for the HBV life cycle. A polyclonal ATP5B KD cell line was established in the NTCP/G2 cells with an shRNA-expressing lentivirus vector (Figure 3A). The knockdown level of ATP5B was confirmed by immunoblot without affecting ATP5A1 expression (Figure 3A). These ATP5B KD cells were infected with HBV, and monitored HBV-related antigens in the soup, such as HBeAg and HBsAg, which are the representative markers to show that HBV infects and produces its related antigens. Then, the respective HBeAg and HBsAg production in the soup was monitored HBeAg every 3 days until 9 days post-infection. The results clearly showed the relationship between the ATP5B expression level and HBV infectivity—i.e., lower ATP5B expression was correlated with lower production of HBeAg and HBsAg.

### 2.6. ATP5B Is Required for HBV Attachment/Entry

Low expression of HBeAg should be caused by several steps of the HBV life cycle; i.e., from the attachment/entry to transcription steps. In order to discriminate what step of the HBV life cycle in the infected cell was disturbed, we attempted to determine the quantity of cccDNA—which was a crucial molecule for HBV gene expression and replication—that was generated after infection. Using ATP5B KD cells for infection, total cellular DNA was prepared to assess how much HBV DNA per cell was incorporated, and cccDNA was quantified to determine much HBV was needed to establish the infection. cccDNA was prepared by plasmid-safe DNase I treatment (see Materials and Methods). As shown in Figure 3D, both total HBV DNA and cccDNA were clearly reduced in a manner dependent on the ATP5B expression level. Because the total HBV DNA could have included HBV DNA in the particles remaining on the cell surface, it could not be taken to represent the real amount of infected intracellular HBV DNA. Compared with the level of reduction of cccDNA with total HBV DNA, the cccDNA formation process seemed to be strongly impaired, and cccDNA formation reflected the number of HBV that entered into the cell. Thus, these results suggested that ATP5B should play some role in the HBV attachment/entry process.

### 2.7. ATP5B Interacts with myrPS1:2-47 at the Front of the ATP-Binding Domain

The above experiments showed that ATP5B was a necessary factor for the HBV attachment/entry process. Next, therefore, we tried to determine which region of ATP5B interacted with myrPS1:2-47. Initially, we tried to express all full-length ATP5B and its deletion mutants in *E. coli*, but some of them could not be expressed well in *E. coli*. Three constructs—full-length ATP5B (1-529aa), 1-198aa and 163-529—were successfully expressed in *E. coli* and the other deletion mutants; however, the 1-275aa, 1-397aa, and 199-266aa regions were not well purified from *E. coli*, and thus we tried to express these regions in mammalian cultured cells as Halo-tag fused constructs. As shown in Appendix A, the full-length ATP5B and 163-529aa regions are clearly bound with the myrPS1:2-47 peptide, and the 1-198aa region is scarcely bound with the myrPS1:2-47 peptide. The results suggested that myrPS1:2-47 could bind directly with ATP5B; that the major binding region of ATP5B with PS1 could be in the 199-529aa region; and that the boundary region around 199aa might be involved in the interaction, since a band could be detected in the bound fraction of ATP5B:199-266aa, though it was very weak. Considering the data obtained from a mammalian expression system (Appendix A), the binding region could be in the 199-266aa region, which covered the N-terminal ATP binding domain of ATP5B (Appendix A).

### 2.8. An ATP5B Regional Peptide Blocks HBV Infection

We then prepared four synthetic peptides, 160-21aa, 195-240aa, 215-250aa, and 225-270aa, of ATP5B corresponding to the interacting region and performed an HBV infection experiment as described above. Although the interaction region analysis suggested that either the 195-240aa or 215-250aa peptide should show the strongest infection inhibition activity, the activity of the peptides was not very strong, and the 160-210aa peptide exhibited stronger inhibitory activity than the other peptides (Appendix A). It was not clear why the putative major interacting region did not show a typical infection inhibition activity, although several possible reasons can be proposed—e.g., it may be that such peptides could not form the correct 3D structure and/or that the peptides were not long enough to inhibit the interaction between HBV and ATP5B and so on.

### 2.9. ATP5B Does Not Interact with NTCP

Based on the above, we considered it was important to determine how ATP5B was involved in HBV infection. The association of ATP5B with NTCP appeared to be important in this context. Thus, we tested whether ATP5B could interact physically with NTCP by using the transient expression of NTCP V5His in HEK293T cells, since the efficiency of transfection of NTCP V5His into HepG2 or the other human HCC-derived cell lines was extremely poor, and there was no suitable antibody to detect NTCP in Western blot analysis. As shown in Figure 4, NTCP V5His expression was fine and was detected clearly in the bound fraction, though it was not detected in the input lane (Figure 4A). It was not clear why the expression was too low to detect in the whole cell lysate (input). In contrast, ATP5B was not detected in the bound fraction but was detected in the unbound fraction (Figure 4B). These data suggested that ATP5B1 and NTCP should not interact physically, at least when there was no HBV.

## 3. Discussion

For many years, the receptor molecules involved in the cellular entry of HBV had been a mystery, making it impossible to observe the HBV life cycle in vitro. With the discovery of NTCP [5], it became possible to render human hepatoma cell lines such as HepG2 competent for HBV infection, although this system was not perfect, because it relied on immortalized cell lines originated from human hepatocellular carcinomas. Nevertheless, it has become a very convenient system of HBV infection to investigate the HBV life cycle and explore anti-HBV compounds, even though human primary hepatocytes have become more available from the livers of humanized mice [23].

Following the discovery of NTCP, several other factors and/or cofactors involved in HBV attachment/entry were found, such as glypican [24], a heparan sulfate proteoglycan [25], and epidermal growth factor receptor (EGFR) [26]. We independently tried to identify such molecules by means of a pull-down assay using myristoylated preS1:2-47, a synthetic peptide, which we suspected was a viral ligand to cellular receptors to HBV [13,27]. Our results showed that ATP5B was constantly pulled down with the peptide in the absence of NTCP (Figure 1 and Appendix A), which suggested that ATP 5B should be a factor interacting with preS1 independent of NTCP. In addition, the peptide did not succeed in pulling down ATP5A1. Since ATP5A1 is also a component of the F_0_F_1_ ATP synthase, the preS1 binding with ATP5B might probably disrupt the complex so that only the ATP5B is pulled down (Figure 1A).

In addition, we could not show that ATP5B interacted with NTCP (Figure 4), suggesting that ATP5B should be an independent factor from NTCP. In this study, we used tagged NTCP at the C-terminus and someone might be afraid that the structure and/or the function were impaired by the tag. In the recently published structural analysis of NTCP, tagged NTCP at the N- and/or C- terminus was used for the expression and purification [28,29]. The report by Park et al. showed myristoylated preS1: 2-47 binding and the HBV infectivity with such construct, and thus tagging itself could not be a major problem.

ATP5B is a b subunit of F_0_F_1_ ATP synthase, H^+^ transporting F_1_ complex, and is basically present at the mitochondrial inner membrane [12,30]; however, the molecule has also been found at the cell surface in multiple studies [14,16,20,21] and has been reported to function as a receptor for angiostatin [16] and apolipoprotein A-I [14]. In addition, some ATP synthases, including ATP5B, were reported to be involved in virus entry and/or the life cycle [8,9,10,11,31,32]. We also confirmed that some ATP5B was present on the cell surface and could interact with the preS1 region of the HBV large S protein on the cell surface (Figure 1 and Figure 2), though our in vivo interaction analysis (IFA) was not sensitive enough to visualize the interaction on the cell surface (Figure 1C). In addition, myristoylation of the region was important for the binding (Figure 1B) and so was asparagine at position 9. Myristoylation of preS1 seemed to also be responsible for the interaction with NTCP [33]. The preS1:2-47 (myrPS1:2-47) seemed to interact with ATP5B through the 199-266 aa region of the protein (Appendix A), as well as some part of the 1-198 aa region, because a band was slightly detected in the 1-198 aa bound fraction (Appendix A). An infection-blocking assay using synthetic peptides (ATP5B: 160-210 aa) around this region showed more efficient inhibition of the infection, further suggesting that the HBV preS1 region should interact with ATP5B on the cell surface to allow entry into the cells (Appendix A). In this context, we should remember that myrcludex B (MyrB) efficiently blocks HBV entry by binding with NTCP [33,34,35]. MyrB is similar to myrPS1 and inhibits bile acid uptake into hepatocytes as well [36]. Combined with our present data, these facts suggest that MyrB could also bind cell surface ATP5B to block HBV entry.

The preS1 region of the large S protein should have a chance to interact with ATP5B at the mitochondria [37], since the molecule was basically a factor residing at the mitochondria, as mentioned above. In our case, however, knockdown of ATP5B led to a reduction in antigen expression concomitant with a reduction in cccDNA formation (Figure 3) and interestingly HBV infectivity was dependent on the level of ATP5B expression (Appendix A). This suggested that some ATP5B on the cell surface, rather than ATP5B in the mitochondria, should be involved in the HBV attachment/entry process on the cell surface.

It is very difficult to evaluate whether the attachment is reduced or not because there is a lot of non-specific binding to the cells; however, taking into consideration that ATP5B is on the cell surface and cccDNA formation (entry) is reduced, we speculate that attachment of HBV on the cell surface is decreased.

In addition, myristoylation of the region was important for the binding (Figure 1B) and so was asparagine at position 9. Myristoylation of preS1 seemed to be also responsible for the interaction with NTCP [33]. The preS1:2-47 seemed to interact with ATP5B through the 199-266 aa region of the protein. An infection blocking assay using synthetic peptides around this region showed that the front side of the binding region more effectively inhibited the infection, suggesting further that the HBV preS1 region should interact with ATP5B on the cell surface to enter into the cells. In this context, it is worth remembering that myrcludex B (MyrB) efficiently blocks HBV entry by binding with NTCP [33,34,35]. MyrB has also been shown to inhibit bile acid uptake into hepatocytes [36]. Combined with our present data, these facts indicate that MyrB could bind cell surface ATP5B to block HBV entry.

Many inhibitors against ATP synthase have been reported [30]. Omeprazole has been reported as Helicobacter pylori F1-ATase [38]. 17-β-estradiol has also been reported as an inhibitor of rat brain mitochondrial F_0_F_1_ ATP synthase. We tested several ATP synthase inhibitors, including omeprazole and 17-β-estradiol, to determine whether they blocked HBV infection. The results showed that these inhibitors did not constantly inhibit HBV infection, suggesting that the enzymatic reactions involved in ATP synthesis and hydrolysis are likely independent (data not shown).

On the other hand, suramin, which was first identified as an inhibitor against human immunodeficiency virus 1 (HIV-1) reverse transcriptase [39], is also known as one of the HBV polymerase inhibitors [40]. Suramin also functions as an entry inhibitor against HBV [41], though its clinical use for chronic HBV infection treatment was abandoned because of its toxicity. It is, however, a very interesting fact that suramin impairs purinergic receptor function and inhibits entry of hepatitis D virus (HDV) as well as HBV [20,41]. Cell surface F_0_F_1_ ATP synthase might be involved in HBV entry along with the purinergic receptor pathway [42] rather than NTCP (Figure 4).

Taken together, our results suggested that ATP5B on the cell surface is involved in HBV attachment and entry into the cell independent of NTCP. Thus, compounds that inhibit the physical interaction between cell-surface ATP5B and HBV could be therapeutic agents against HBV infection, although further study is needed.

## 4. Materials and Methods

### 4.1. preS1 Peptide Pulldown

A myristoylated preS1 peptide (2-47aa: GTNLSVPNPLGFFPDHQLDPAFGANSHNPDWDFNPNKDHWPEANQV) (myrPS1) was synthesized by a manufacturer (Scrum); its N-terminal glycine was myristoylated and its c-terminus was linked with lysine-biotin and then amidated. The non-myristoylated version of the preS1 peptide and the mutant versions at N9K and F13D, respectively, were also synthesized. A PreS2 peptide (2-55aa: QWNSTTFHQALLDPRVRGLYFPAGGSSSGTVNPVPTTASPISSISSRTGDPAP) was also synthesized in the same way. The peptides were adjusted to 0.1 µM in DMSO and bound with streptavidin-Sepharose^®^ (GE−Healthcare, Chicago, IL, USA). The cell lysate of HepG2 cells or NTCP-expressing HepG2 cells was prepared in a lysis buffer (20 mM Tris-HCl, pH 8.0, 0.3 M NaCl, 1 mM 2-mercaptethnol, 0.1% Triton-X100™ and a protease inhibitor for the mammalian cell lysate; Sigma, St. Louis, MO, USA). The protein concentration was adjusted to 1 mg/mL and the 10 mL was incubated with the preS1 peptide-bound Sepharose overnight at 4 °C and then centrifuged to pellet the Sepharose. The Sepharose was washed three times with the lysis buffer and then suspended in 150 µL of 1× sample buffer (62.5 mM Tris-HCl, pH 6.8, 1% SDS, 10% glycerol, 0.02% bromophenol blue, and 70 mM 2-mercaptoethanol) and boiled for 10 min. A control sample that bound with streptavidin-Sepharose only was prepared in the same way. The extracted samples were separated on an SDS-PAGE and stained with SyproRuby^®^ (BioRad, Hercules, CA, USA) according to the manufacturer’s instructions. Several specific bands for the myrPS1 peptide were subjected to MALDI-TOF/MS analysis and an attempt was made to identify the proteins.

### 4.2. Cells

HepG2 cells and their derivatives (NTCP-expressing HepG2 cells [NTCP/HepG2]) were maintained in a Williams’ B culture medium (Gibco)-based PMM (primary hepatocyte maintaining medium), which contained 10% fetal bovine serum (FBS), 100 IU/mL penicillin, 100 µg/mL streptomycin and 0.25 µg/mL amphotericin B (Nacalai Tesque, San Diego, CA, USA), 50 µM hydrocortisone (Sigma-Aldrich), 5 µM dexamethasone, 5 µg/mL transferrin (Wako Pure Chemicals), 10 ng/mL EGF (ThermoFisher, Waltham, MA, USA), 5 µg/mL insulin (Sigma-Aldric), 5 ng/mL sodium selenite, 2 mM L-glutamine (Nacalai Tesque), and 0.5 mg/mL G418 (Nacalai Tesque). NTCP-expressing HepG2 cells were the kind gift of Dr. Watashi [43,44] and were maintained in the same PMM but with the addition of 0.5 mg/mL G418 (Nacalai Tesque). HEK293T cells were cultured in Dulbecco’s modified Eagle media (DMEM) (high glucose) (Nacalai Tesque) supplemented with 10% FBS, 100 IU/mL penicillin, 100 µg/mL streptomycin, and 0.25 µg/mL amphotericin B (Nacalai Tesque). All the cells were maintained in a 5% CO_2_ incubator (Panasonic Health Care, Tokyo, Japan). Huh6 and HB611 cells were maintained in DMEM (low glucose) (Nacalai Tesque) with the same supplements as HEK293T. In the case of HB611, G418 (Nacalai Tesque) was added to the medium at 0.5 mg/mL.

### 4.3. Plasmids

A pLKO-based knockdown vector that expressed shRNA was purchased from a manufacturer (Sigma-Aldrich). The shRNA sequence used for ATP5B knockdown was 5′-CCGGCACAGTAAGGACTATTGCTACTCGAGTAGCAATAGTCCTTACTGTGCTTTTTG-3′ (TRCN0000043437). TRC2-pLKO-puro non-target shRNA#1 (SHC202) was used as a control vector.

NTCP tagged with V5-His_6_ was initially constructed in a pMT/V5-His vector (Invitrogen, Waltham, MA, USA) and then the NTCP V5-His fragment was cloned into the EcoRI site of a pQc XIN vector (Takara-Clontech, Kusatsu, Japan) and termed pQc XIN NTCP V5-His.

### 4.4. Knockdown of ATP5B

In total, 1,000,000 Tet293 cells (Takara-Clontech) were seeded on a collagen-coated 10 cm dish (AGC Techno Glass) 1 day before transfection. The next day, an shRNA against the ATP5B expression vector (7 µg) was transfected into Tet293 cells (Takara-Clontech) with a packaging mix (Lenti-X Packaging Shots [VSV-G]) (Takara-Clontech) according to the manufacturer’s instructions. At 7–8 h after transfection, the medium was refreshed and the cells were incubated for an additional 2 to 3 days. Then, the medium was harvested and passed through a 0.45 µm filter (Millipore, Burlington, MA, USA). The lentivirus stock was contacted with 10^5^ of NTCP/G2 cells in a 3 cm dish overnight, and then the medium was refreshed and the cells were incubated for an additional 2 days. The cells were then passed into two 10 cm dishes (AGC Techno Glass) in 0.5 mg/mL G418 and 5 µg/mL puromycin-containing media for selection. The medium was changed every 3 days and when colonies were visible, expression of ATP5B was screened as a polyclonal clone and polyclonal ATP5B KD cells were established.

### 4.5. Cell Surface Labeling

Sulfo-NHS-LC-LC-biotin (Thermo Scientific) was added to the medium of HepG2 or NTCP-expressing HepG2 cells for 30 min. The medium was withdrawn and the cells were washed with PBS (-) three times and then lysed in a lysis buffer (50 mM NaPO4-0.3 M NaCl-0.1% Triton^®^-X100). After excluding the cell debris, the lysate was incubated with streptavidin-Sepharose to pull down. The unbound fraction was obtained by centrifugation and the bound fraction was washed with the lysis buffer three times and eluted with 1× sample buffer (62.5 mM Tris-HCl, pH 6.8, 70 mM 2-mercaptoethanol 1 mg/mL bromo-phenol blue, 10% glycerol and 2% SDS).

### 4.6. HBV Infection Experiment with NTCP/G2

An HBV infection experiment was performed using NTCP/G2 or ATP5B knocked-down cells. The cells were seeded 1 day before infection at 5 × 10^5^/well (collagen-coated 24-well plate; AGC Techno Glass, #4820-010), and a 1000 genome equivalent of infection (GEI) of HBV was contacted with the cells in 4% PEG 8000 (Sigma)-2% DMSO-PMM for 1 day. The next day, the cells were washed with 1 mL of 2% DMSO-PMM three times and then fresh 1 mL of 2% DMSO-PMM was added to each well (day 0). Sampling (100 µL) of soup from the infected cells was usually conducted at day 0 and then every 3 days until 9 days post-infection (the last day). HBeAg in the soup was measured with an e Antigen ELISA Kit (Bioneovan Co., Beijing, China).

HBV was prepared from HepAD38.7 cells, in which HBV production was controlled by a TetOff system [45]. The supernatant of HepAD38.7 cells induced for HBV production by the withdrawal of tetracycline was accumulated and subjected to 10% PEG8000 precipitation. The precipitate was dissolved in phosphate-buffered saline (PBS), and unsolved materials were removed by passing through a 0.45 µm filter (Millipore, Cat. SLHV033RB). The initial volume of PBS used for solubilization was finally expanded to an at-least 10-fold volume to exclude PEG as much as possible, and then concentrated again with a 100 kd filter.

### 4.7. HBV Genome Quantification

Particle-associated HBV DNA was prepared from the 100 µL final PEG precipitated preparation for the quantification analysis. Briefly, DNA and RNA outside of the particle were degraded with DNase I (Takara-Clontech) and RNase at 37 °C for 30 min. DNase I was inactivated by adding EDTA, pH 8.0, at 10 mM and incubating at 70 °C for 30 min. Then, particle-associated HBV DNA was extracted with 0.2 mg/mL protease K (Roche) in the presence of 1% SDS overnight or for more than 8 h incubation at 56 °C. Ten micrograms of sonicated salmon sperm DNA was added and extracted with phenol-chloroform-isoamylalcohol (24:24:1) (PCI), and finally the DNA was precipitated with ethanol and with 0.25 mg/mL glycogen (Nacalai Tesque), and then dried and dissolved in 100 µL TE (10 mM Tris-HCl, pH 7.8, 1m M EDTA, pH 8.0).

HBV DNA in 1 µL of the solution was quantified with a QuantStudio^TM^ Real-Time PCR System using Fast SYBR^TM^ Green Master Mix (Applied Biosystems-Thermo Fisher scientific, Cat. #4385612) using the following primer set: qPCR primers for HBV DNA (forward, 5′-CTTCATCCTGCTGCTATGCCT-3′; reverse, 5′-AAAGCCCAGGATGATGGGAT-3′) and QuantStudio 6 Flex (Applied Biosystems by Life Technologies). The PCR conditions consisted of an initial cycle of 95 °C for 20 s, followed by 40 cycles of 95 °C for 1 s and 60 °C for 20 s, with a final cycle of 95 °C for 15 s, 60 °C for 1 min, 95 °C for 30 s, and 60 °C for 15 s. The HBV DNA copy number was measured according to the standard HBV copy number.

For cccDNA quantification, total DNA was extracted from the infected and non-infected NTCP/G2 cells on the designated day. Typically, the DNA was prepared on the day of infection, at the end of the infection, and on days 3, 6, and 9 after infection. The amount of total DNA was measured and 30 µg of the DNA was treated with Plasmid-Safe™ ATP Dependent DNase (EpicenterBio) according to the manufacturer’s instructions. The DNA was extracted with PCI as described above, and then precipitated with ethanol in the presence of 0.25 mg/mL glycogen (Nacalai Tesque), followed by rinsing with 70% ethanol and drying. Finally, the dried pellet was suspended in 10 µL TE. One microliter of the sample, which corresponded to 3 µg total DNA, was subjected to qPCR with primers specific to cccDNA as reported previously (ref). The sequences were 5′-GTCTGTGCCTTCTCATCTGC-3′ (Fw) and 5′-GCACAGCTTGGAGGCTTGAA-3′ (Rv). Quantification was performed using the method described above.

### 4.8. Transfection and Pull-Down of NTCP V5-His

One day before transfection, 2 × 10^6^ 293T cells were seeded on a 10-cm dish (collagen-coated; Iwaki Glass™). The next day, ten µg pQc XIN NTCP V5-His was transfected to the cells with LT-I^®^ transfection reagent (Takara-Clontech) according to the manufacturer’s instructions. Two days after transfection, cells were harvested and lysed in a Nickel-NTA (Ni-NTA) buffer (20 mM NaPO_4_, pH 8.0, 0.3 M NaCl, 10% glycerol, 10 mM 2-mercaptoethanol, and 0.1% Triton^®^ X-100). The cell debris was clarified with centrifugation and the cleared lysate was passed through a 0.22 µm filter unit (Millipore). The obtained lysate was mixed with Ni-NTA agarose and incubated with agitation at 4 °C overnight. The Ni-NTA agarose was washed with the same buffer three times, and the bound fraction was eluted with 0.5 M imidazole.

### 4.9. Western Blot

Western blotting analysis was performed with ordinary methods. Briefly, protein samples were separated on SDS-PAGE and transferred onto a piece of PVDF membrane (Bio Rad). The membrane was blocked with a 10% dry milk-containing TBS T buffer (20 mM Tris-HCl, pH 7.6, 137 mM NaCl, and 0.1% Tween 20) for 30 min at RT (room temperature) and washed with TBS-T for 5 min three times at RT. Then, the membrane was reacted with appropriate primary antibodies followed by washing with TBS and then incubation with secondary antibodies conjugated with horseradish peroxidase (HRP). The signal was detected as luminescence with a chemiluminescence-generating kit (Super Signal™ West Pico Chemiluminescent Substrate; Fisher Scientific) and an imager (ChemiDoc™ Touch MP imaging system; Bio Rad) according to the manufacturer’s instructions. Otherwise, the membrane was blocked with an Odyssey™ (LI-COR Bioscience) blocking buffer (0.1% bovine serum albumin [BSA], 0.1% gelatin, and 0.01% sodium dodecyl sulfate [SDS] in PBS). Incubation with antibodies was performed in this buffer and then the membrane was washed with PBS-T four to five times for 5 min each at RT. The membrane was then incubated with secondary antibodies: donkey anti-mouse IgG (H+L) antibodies or donkey anti-rabbit IgG antibodies conjugated either with IRDye^®^ 680LT or 800CW (LI-COR^®^) according to the orientation. Finally, the membrane was washed with PBS-T five times for 5 min each, followed by two washings in PBS for 5 min each. The signal was obtained with an infrared imaging system (Odyssey™, LI-COR).

### 4.10. Immunofluorescence Assay (IFA)

Huh6 or HB611 cells were seeded at 5 × 10^4^/well (8 chamber slide; Matsunami Glass). After confirming that the cells were attached, the cells were fixed with 4% paraformaldehyde in PBS (-) overnight at 4 °C and permeabilized with 0.1% TritonX-100 in PBS (-) for 30 min at RT, followed by washing with water. After dehydration with 70%, 80%, 100%, and 100% ethanol for 5 min each and drying, the cells were probed with rabbit anti-ATP5B polyclonal antibodies (Sigma-Aldrich) and a mouse anti-preS1 monoclonal antibody (Beacle Inc.), diluted in PBS-T containing 0.25% BSA and 0.02% Na_3_N and incubated at RT overnight. Then, the cells were washed three times with PBS-T, and probed with the secondary Alexa Fluor^®^ 488-conjugated goat anti-mouse and Alexa Fluor^®^ 546-conjugated goat anti-rabbit antibodies for 3 h at RT. After washing with PBS-T three times, the cells were dehydrated with 70%, 80%, 100%, and 100% ethanol for 1 min each and air-dried. Finally, the cells were mounted with a DAPI (4’,6-diamidino-2-phenylindole)-containing glycerol solution (Fluoro-KEPPER Antifade Reagent; Nacalai Tesque) and the image was obtained with a laser scanning confocal microscope (Leica TCS SP8) with an x 63 objective lens and with 3-fold zooming.

### 4.11. Primary Antibodies

The following primary antibodies were purchased from the manufacturers: ATP5A1 (Sigma-Aldrich, SAB4502040), ATP5B (Sigma, HPA001520), GAPDH (Sigma-Aldrich, G8795), b-tubulin (Sigma-Aldrich, T5201), HaloTag^®^ (Promega, Madison, WI, USA, G921A), and V5 (Nacalai Tesque, V5005).

### 4.12. Statistical Analysis

Data are presented as the mean ± standard deviation of at least three independent experiments. Student’s *t*-test was used to evaluate the statistical significance of differences, where *p* < 0.05 was considered significant. The asterisks on the figures represent * < 0.05, ** < 0.01, and *** < 0.001.

## 5. Conclusions

ATP5B exists on the cell surface and is involved in HBV entry.

## Figures and Tables

**Figure 1 ijms-23-09570-f001:**
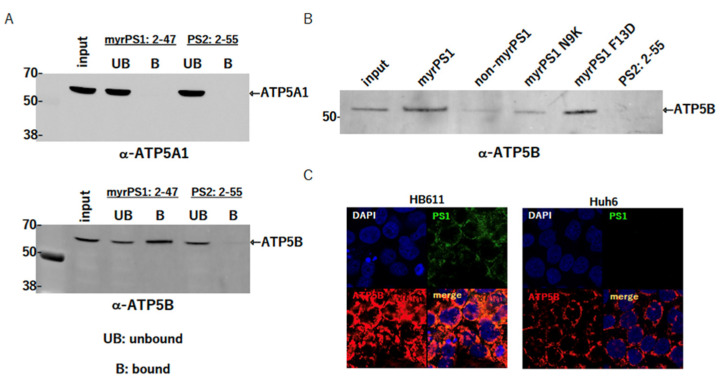
preS1 or preS2 peptide pull-down assay. (**A**). Immunoblot assay of pulled-down samples. Pulled-down samples with myristoylated preS1: 2-47 or preS2:2-55 were blotted either with an anti-ATP5A1 (**upper**) or an anti-ATP5B antibody (**bottom**). (**B**). Immunoblot assay of several kinds of mutant preS1 peptides and preS2 peptides. Pulled-down samples were blotted with an anti-ATP5B antibody. PS1, preS1; PS2, preS2; UB, unbound fraction; B, bound fraction; myr, myristoylated. (**C**). Cellular localization of ATP5B and preS1 (LS). HB611 and Huh6 cells were double-stained with the anti-ATP5B antibody (rabbit) and an anti-preS1 antibody (mouse) followed by Alexa Fluor^®^ (LI-COR Biosciences, Lincoln, NE, USA) 488-conjugated goat anti-mouse and Alexa Fluor^®^ 546-conjugated goat anti-rabbit antibodies (LI-COR Bioscicences, Lincoln, Nebraska). Nucleus (DAPI), preS1 (green), ATP5B (red), and the merged images (merge) are shown. The original images were obtained with an ×63 objective lens and with 3-fold zooming.

**Figure 2 ijms-23-09570-f002:**
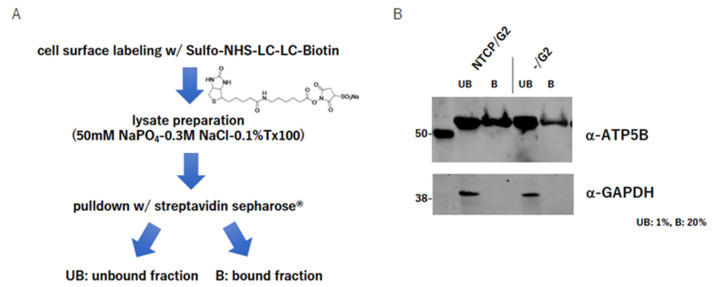
Detection of ATP5B on the cell surface. (**A**) The method of cell surface labeling. Sulfo-NHS-LC-LC-biotin was added to the medium of HepG2 or NTCP-expressing HepG2 cells for 30 min. The medium was withdrawn and the cells were washed with PBS (-) three times, and then lysed in a lysis buffer (50 mM NaPO_4_-0.3 M NaCl-0.1% Triton^®^-X100 (Sigma-Aldrich, Darmstadt, Germany)). After excluding the cell debris, the lysate was incubated with streptavidin-Sepharose^®^ to pull down. The unbound fraction was obtained by centrifugation and the bound fraction was washed with the lysis buffer three times and eluted with 1× sample buffer (62.5 mM Tris-HCl, pH 6.8, 70 mM 2-mercaptoethanol 1 mg/mL bromo-phenol blue, 10% glycerol and 2% SDS). (**B**) Detection of cell surface labeled protein. The boiled samples in the 1× sample buffer were separated on the SDS-PAGE, blotted on a PVDF membrane and probed either with an anti-ATP5B antibody (**upper**) or an anti-GAPDH antibody (**bottom**).

**Figure 3 ijms-23-09570-f003:**
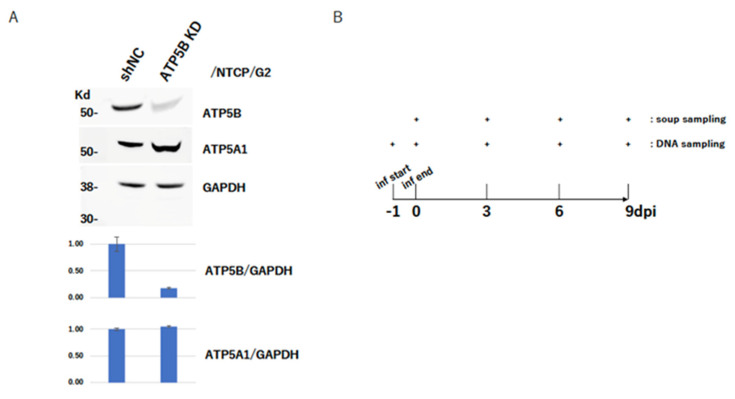
Low efficiency of HBV infection to ATP5B−knocked down cells. (**A**) Establishment of ATP5B cells in NTCP−expressing HepG2 cells. ATP5B and ATP5A1 expression levels were confirmed by Western blotting analysis. GAPDH was also evaluated as a loading control. The graphs under the blots are the expression (band intensity) ratio of either ATP5B or ATP5A to that of GAPDH. The data are shown as representatives. (**B**) Sampling method of the infection experiment. Cells were prepared 1 day before infection, and the next day the virus started to come into contact with the cells (−1 dpi). The virus was withdrawn and the cells were washed (0 dpi), and then every 3 days up to 9 dpi the medium was refreshed and the samples (soup and/or DNA) were harvested or prepared as shown. (**C**) sAg ELISA of the infection experiment. Obtained ELISA data of OD_450-630_ are shown as mean values with the standard deviation. (**D**) eAg ELISA of the infection experiment. Obtained ELISA data of OD_450-630_ are shown as mean values with the standard deviation. (**E**) Quantification of intracellular HBV DNA in the infection experiment. Total DNA was extracted on the indicated day and 10 ng DNA was used for quantification of the HBV DNA copy number. The data are shown as mean values with the standard deviation. (**F**) Quantification of intracellular cccDNA in the infection experiment. Total DNA was extracted on the indicated day and 10 µg DNA was treated with Plasmid-Safe™ ATP Dependent DNase as described in the Materials and Methods. After inactivating the enzyme, the DNA was used for quantification of the HBV DNA copy number. The data are shown as a mean value with the standard deviation. Statistic values were evaluated on the final day, where two groups of data were compared. ** *p* < 0.01.

**Figure 4 ijms-23-09570-f004:**
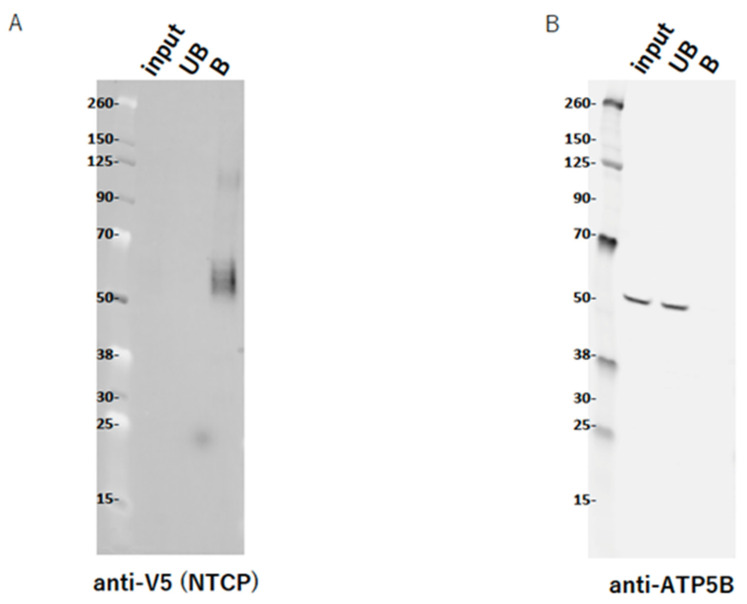
NTCP and ATP5B interaction assay. An NTCP V5-His expression vector was transfected to 293T cells. Total cell lysate was prepared 2 days after transfection, and the expressed NTCP V5-His was purified with Ni-NTA. The input, unbound (UB), and bound (B) fractions were electrophoresed on the SDS-PAGE and analyzed with Western blotting analysis for NTCP (**A**, anti-V5) and ATP5B (**B**).

## Data Availability

All data generated or analyzed during this study are included in the published article.

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
