# Peer review of "ATP5B Is an Essential Factor for Hepatitis B Virus Entry"

_ijms, 2022, doi:10.3390/ijms23179570_

Round 1

Reviewer 1 Report

1) There is little background description of HBV. What family is HBV from? Is it a DNA virus?  How is it spread?  The following information should be included:

"HBV, a member of the Hepadnaviridae family, is a small DNA virus with unusual features similar to retroviruses. HBV replicates through an RNA intermediate and can integrate into the host genome. The unique features of the HBV replication cycle confer a distinct ability of the virus to persist in infected cells." From the web

https://www.ncbi.nlm.nih.gov/pmc/articles/PMC2809016/

2) While the experiment is ok as far I can tell, some of the sentences used are awkward and hard to understand. I can understand this since English is obviously not the first language of the authors.  Examples are:

A)

"HBV itself was found more than a half 26century ago [2, 3], but there had been none to allow HBV to infect in vitro and thus detail 27analysis of the HBV life cycle had been extremely hampered"

There should be a comma before "and thus".

I am also not sure what the authors were trying to say, I think that they were trying to say, "There so few cases to conduct HBV experiments in vitro"

B)

" Some of them will work independently of or dependently 39on NTCP."

should be corrected to:

" Some of them will work independently of or are dependent on NTCP."

C)

"One of such issues should be whether NTCP were an only factor required for HBV 36attachment and entry into the cells and there should be many other cofactors for the HBV 37entry as proved for the other virus entry like human immunodeficiency virus 1 (HIV-1) 38and hepatitis C virus (HCV)."

NTCP is or was, "an only" should be "the only" . There should be a comma between "cells" and "and".

3) I noticed that the authors like to use long sentences. They should  try tpo break down the long sentences to multiple sentences. Otherwise, it is easy to make grammatical errors or the sentence can easily become very confusing.

Author Response

We would like to thank for comments and suggestions by reviewers.

Basically, English was proof-read by native speakers. In addition, some description was corrected. For example, ATP5A should be ATP5A1 and ATP5B1 should be ATP5B.

The corrected sentences are underlined.

Reviewer 1

1) There is little background description of HBV. What family is HBV from? Is it a DNA virus? How is it spread? The following information should be included:

"HBV, a member of the Hepadnaviridae family, is a small DNA virus with unusual features similar to retroviruses. HBV replicates through an RNA intermediate and can integrate into the host genome. The unique features of the HBV replication cycle confer a distinct ability of the virus to persist in infected cells." From the web

https://www.ncbi.nlm.nih.gov/pmc/articles/PMC2809016/

>> It has been added at the introduction with some modification.

2) While the experiment is ok as far I can tell, some of the sentences used are awkward and hard to understand. I can understand this since English is obviously not the first language of the authors.  Examples are:

  1. A) "HBV itself was found more than a half century ago [2, 3], but there had been none to allow HBV to infect in vitro and thus detail analysis of the HBV life cycle had been extremely hampered"

There should be a comma before "and thus".

I am also not sure what the authors were trying to say, I think that they were trying to say, "There so few cases to conduct HBV experiments in vitro"

>> l31-32: It has been corrected to "HBV itself was found more than a half century ago [2, 3], but there had been none to allow HBV to infect in vitro, and thus there had been so few cases to conduct HBV experiments in vitro."

  1. B) "Some of them will work independently of or dependently on NTCP."

should be corrected to: "Some of them will work independently of or are dependent on NTCP."

>> l43: It has been corrected as pointed out.

  1. C) "One of such issues should be whether NTCP were an only factor required for HBV attachment and entry into the cells and there should be many other cofactors for the HBV entry as proved for the other virus entry like human immunodeficiency virus 1 (HIV-1) and hepatitis C virus (HCV)."

NTCP is or was, "an only" should be "the only". There should be a comma between "cells" and "and".

>> l41: It has been corrected as pointed out.

3) I noticed that the authors like to use long sentences. They should try tpo break down the long sentences to multiple sentences. Otherwise, it is easy to make grammatical errors or the sentence can easily become very confusing.

>> Thank you for your comment. And I have carefully checked the manuscript and corrected.

Reviewer 2 Report

In this manuscript Ueda and Suwanmanee present findings that identify ATP5B as as essential factor for the HBV entry. They found that ATP5B was expressed on the cell surface of the HCC cell lines and bound with myristoylated preS1 2-47. Further, KD of ATP5B in an NTCP expressing HepG2 cell, which allowed HBV infection, reduced HBV infectivity with less cccDNA formation. Taken together, these results strongly suggest that ATP5B should be an essential factor for HBV entry.

Overall, findings presented are interesting. However, several points need to be addressed before the manuscript can be recommended for publication.

Figure 1C is of too low resolution to be evaluated effectively. Better resolution images would be helpful here. I also noticed that the authors have permeabilized the cells during the IFA protocol. If the staining was repeated without permeabilization, it may improve chances of adequate staining of surface antigens.

In Fig 3E, Considering, ATP5B may be an attachment factor rather than entry, the knockdown cells may just have less particles bound and thus less particles may initially enter the cells and explain the effect seen in 3E. I think an experiment to delineate whether the inhibition is due to reduced cell attachment vs reduced entry would be helpful.

Throughout the text descriptions of experimental design as well as implications of the research findings feel a little under-explained. Providing greater context while during the description of the experiment itself will greatly help non-specialists to completely grasp of important technical details and/or it’s implications. For example,

-Line 66: What is the implication of myristyolated pre-S1 in section 2.1? This is not explained until section 2.2.

-Line 129:131- what is the reason to measure HBsAg and HBeAg? That is explained partially in line 134, but what about HBsAg?

-Also, line 204, why was there no band in the input lane? It is likely that the protein was below the limit of detection in the lysate, but it would be helpful if authors explained that in the text.

Other minor points:

The text mentions supplementary figures, but these are not provided with the document, only original blots are available.

Line 62: The formatting instructions should be deleted.

Line: 251 “also” instead of “alos”

Overall, I hope that these comments help the authors in strengthening their manuscript and ensuring a successful publication.

Author Response

We would like to thank for comments and suggestions by reviewers.

Basically, English was proof-read by native speakers. In addition, some description was corrected. For example, ATP5A should be ATP5A1 and ATP5B1 should be ATP5B.

The corrected sentences are underlined.

Reviewer 2

In this manuscript Ueda and Suwanmanee present findings that identify ATP5B as as essential factor for the HBV entry. They found that ATP5B was expressed on the cell surface of the HCC cell lines and bound with myristoylated preS1 2-47. Further, KD of ATP5B in an NTCP expressing HepG2 cell, which allowed HBV infection, reduced HBV infectivity with less cccDNA formation. Taken together, these results strongly suggest that ATP5B should be an essential factor for HBV entry.

Overall, findings presented are interesting. However, several points need to be addressed before the manuscript can be recommended for publication.

  1. Figure 1C is of too low resolution to be evaluated effectively. Better resolution images would be helpful here. I also noticed that the authors have permeabilized the cells during the IFA protocol. If the staining was repeated without permeabilization, it may improve chances of adequate staining of surface antigens.

>> We have improved the figures.

  1. In Fig 3E, Considering, ATP5B may be an attachment factor rather than entry, the knockdown cells may just have less particles bound and thus less particles may initially enter the cells and explain the effect seen in 3E. I think an experiment to delineate whether the inhibition is due to reduced cell attachment vs reduced entry would be helpful.

>> This is a very good question. In our assay, reduction of ATP5B expression led to HBV entry, since cccDNA formation was reduced accordingly. It is very difficult to evaluate whether attachment is reduced or not, because there are a lot of non-specific binding to the cells. However, taking into consideration that ATP5B is on the cell surface and cccDNA formation (entry) is reduced, we speculate that attachment of HBV on the cell surface is decreased. We have added the sentences to the context (line 299-302).

  1. Throughout the text descriptions of experimental design as well as implications of the research findings feel a little under-explained. Providing greater context while during the description of the experiment itself will greatly help non-specialists to completely grasp of important technical details and/or it’s implications. For example,

-Line 66: What is the implication of myristyolated pre-S1 in section 2.1? This is not explained until section 2.2.

>> We added the description to show the importance; "HBV large S envelope protein is myrstoylated at the 2nd amino acidglycine next to the methionine and it is functionally very important for the attachment and entry of HBV into the cells" (line 72-74) with two references.

  1. -Line 129:131- what is the reason to measure HBsAg and HBeAg? That is explained partially in line 134, but what about HBsAg?

>> HBeAg and HBsAg are the products produced from HBV after infection. We have added the explanation, " HBV large S envelope protein is myrstoylated at the 2nd amino acidglycine next to the methionine and it is functionally very important for the attachment and entry of HBV into the cells" (line 151-153).

  1. -Also, line 204, why was there no band in the input lane? It is likely that the protein was below the limit of detection in the lysate, but it would be helpful if authors explained that in the text.

>> We think that expression level of NTCP V5His is always very low with unclear reasons. We chose HEK293T cells to overcome low transfection efficiency in human hepatoma derived cell lines but even in the HEK293T cells, NTCP V5His was too low to detect the expression in the whole lysate (input). We have added the sentences,"It was not clear why the expression was too low to detect in the whole cell lysate (input)."

Other minor points:

  1. The text mentions supplementary figures, but these are not provided with the document, only original blots are available.

>> We have check the figures and provided original pictures.

  1. Line 62: The formatting instructions should be deleted.

>> We should have done so.

  1. Line: 251 “also” instead of “alos”

>> It has been corrected.

Overall, I hope that these comments help the authors in strengthening their manuscript and ensuring a successful publication.

>> Thank you for your suggestions and comments.

Reviewer 3 Report

In this paper, Ueda and Suwanmanee analyzed functions of ATP5B in the entry step of HBV. ATP5B is expressed on the cell surface of hepatocellular carcinoma cell line HepG2. They found an interaction between ATP5B and myristoylated preS1 and knockdown of ATP5B reduced HBV infectivity using NTCP-expressing HepG2 cells. The results are interesting but there are several issues to be addressed as below.

1.     The results of this study based on a hepatocellular carcinoma cell line basically. Please show whether ATP5B knockdown affects HBV infectivity using primary hepatocytes or another cell line.

2.     Figure 1c – the images of fluorescence microscopy are too small and the magnification is too low to see the colocalization of ATP5B and preS1. Please show images with a higher magnification.

3.     Figure 3 – the knockdown of ATP5B also reduced the expression level of ATP5A. Therefore, there is a possibility that the inhibition effect on HBV entry could be via the reduction of ATP5A. To exclude the possibility, it is required to show an effect of knockdown of ATP5A on HBV entry.

4.     The result in Fig. 4 indicates that there is no interaction between NTCP and ATP5B. However, it is hard to judge whether there is really no interaction based on this method, because a tag of NTCP or conformational change of either protein in the used condition might affect the interaction. Please show a result of positive control protein that is known to interact with NTCP.

5.     Page 3, line 103 -- “in vivo” does not fit this sentence. Please consider to modify the sentence.

6.     Page 3, line 106 – “(ref)” need to fix to show a reference number.

7.     I could not access data of Supplementary figures.

Author Response

We would like to thank for comments and suggestions by reviewers.

Basically, English was proof-read by native speakers. In addition, some description was corrected. For example, ATP5A should be ATP5A1 and ATP5B1 should be ATP5B.

The corrected sentences are underlined.

Reviewer 3

In this paper, Ueda and Suwanmanee analyzed functions of ATP5B in the entry step of HBV. ATP5B is expressed on the cell surface of hepatocellular carcinoma cell line HepG2. They found an interaction between ATP5B and myristoylated preS1 and knockdown of ATP5B reduced HBV infectivity using NTCP-expressing HepG2 cells. The results are interesting but there are several issues to be addressed as below.

  1. The results of this study based on a hepatocellular carcinoma cell line basically. Please show whether ATP5B knockdown affects HBV infectivity using primary hepatocytes or another cell line.

>> I agree with this comment and we should have done the experiment. But it costs much and is very hard to do in 10 days.

  1. Figure 1c – the images of fluore

since microscopy are too small and the magnification is too low to see the colocalization of ATP5B and preS1. Please show images with a higher magnification.

>> The 1C was replaced with more magnified images. The images were obtained originally with a x 63 objective lens and with further 3-fold zooming.

  1. Figure 3 – the knockdown of ATP5B also reduced the expression level of ATP5A. Therefore, there is a possibility that the inhibition effect on HBV entry could be via the reduction of ATP5A. To exclude the possibility, it is required to show an effect of knockdown of ATP5A on HBV entry.

>> When we checked ATP5A1 expression in the isolated ATP5B KD clones, ATP5A1 expression was not changed in all clones with variable expression of ATP5B and HBV infectivity was basically dependent on the level of ATP5B expression (see below). Therefore, we retested the expression using freshly prepared samples from the polyclonal clones, and then we confirmed that ATP5A1 expression was not changed. Thus, we replaced the figure and corrected the results and the discussion according to the newest data. The data of the isolated clones were added to the supplementary information as Figure S3.

  1. The result in Fig. 4 indicates that there is no interaction between NTCP and ATP5B. However, it is hard to judge whether there is really no interaction based on this method, because a tag of NTCP or conformational change of either protein in the used condition might affect the interaction. Please show a result of positive control protein that is known to interact with NTCP.

>> In the recently published reports on structural analysis of human NTCP, tagged NTCP was expressed and purified for the analysis (Asami, J. et al., Nature 606: 1021, 2022 and Park, J.-H. et al., Nature 606: 1027). The latter performed infection analysis too. And thus, tagged NTCP should not necessarily lose infectivity for HBV.

  1. Page 3, line 103 -- “in vivo” does not fit this sentence. Please consider to modify the sentence.

>> l109: I have changed to "in the cell".

  1. Page 3, line 106 – “(ref)” need to fix to show a reference number.

>> I have corrected.

  1. I could not access data of Supplementary figures.

>> I am very sorry about this, but I do not know why.

Round 2

Reviewer 1 Report

Improvement seen

Reviewer 2 Report

All my concerns have been addressed adequately. I have no problems with the publication of the manuscript.

Reviewer 3 Report

The authors addressed the comments from reviewers. The manuscript has been improved.